# The Governance Approach of Smart City Initiatives. Evidence from Trondheim, Bergen, and Bodø [†]

**Savis Gohari [1,\*]** , **Dirk Ahlers [1]** , **Brita F. Nielsen [1] and Eivind Junker [2,‡]**

[1]  Department of Architecture and Planning, Norwegian University of Sscience and Technology (NTNU), 7491 Trondheim, Norway; dirk.ahlers@ntnu.no (D.A.); brita.nielsen@ntnu.no (B.F.N.)

[2]  Faculty of Social Sciences, Nord University, 7600 Levanger, Norway; eivind.junker@nord.no

\*  Correspondence: savis.gohari@ntnu.no

†  This paper is an extended version of an earlier conference presentation (International Conference on Smart Cities (ICSC)). It has been modified for publishing in the journal.

‡  Work done while at NTNU.

**Abstract:** A pragmatic and polity-focused solution for governing a smart city in the direction of sustainability is still missing in theory and practice. A debate about whether a smart city is a pragmatic solution for modern challenges or just a technology-led urban utopia is entangled with the vexed issue of governance. While 'smart governance' has drawn unprecedented interest, the combination of its conceptual vagueness and broad applications couple with a lack of focus on its underlying international and local political paradigms have raised concerns about its utility. This study contributes to restoring attention to the original concept of governance, its differences with governing and government, and the potential challenges resulting from its functionality in its real, multi-layered, and complex contexts. This paper explores the intellectual connection between governance and smart cities, from both an empirical and a conceptual/analytical perspective. From the empirical side, we examine which actors, processes, and relational mechanisms at different levels that have had an impact on the initiation of smart cities in three Norwegian cities: Trondheim, Bergen, and Bodø. We illustrate how the structural sources of the interests, roles, and power in smart city initiatives have caused governance to emerge and change, but have also affected the goals designed by specific actors.

**Keywords:** smart city; sustainability; smart sustainable cities; governance; city planning; informality; Trondheim; Bergen; Bodø

## 1. Introduction

The current rates of growth and resource consumption in cities are fundamentally unsustainable. The smart city concept has introduced the assumption that the integration of smart technologies can resolve some of these pressing modern challenges and mitigate the impacts of rapid urbanization [1–4]. In city planning, which has a normative, strategic, and ideological nature [5], the value of the smart city lies in its capacity to reach sustainable development goals through data generation and management [5–12] as supporting technologies. In addition, emphasis is placed on a radical shift in both existing technology and the organization of society [11,13,14]. However, the concepts of smart cities and sustainable development are often used loosely to describe and explain a host of different things, embracing diverse technical, environmental, economic, political, legal, and social aspects [10,15]. There is not yet a full consensus in the literature on which set of phenomena can properly be grouped under the moniker of a smart city, while the needed integration of smart cities and sustainability is becoming more mainstream [3,10,16,17]. However, many scholars agree on the

six main dimensions of a smart city: a smart economy, smart mobility, a smart environment, smart people, smart living, and smart governance [16,18,19]. Castelnovo et al. [2] argue that there is no clear framework for assessing the real level and content of these dimensions of smartness or to determine the extent to which technology can contribute to improving the quality of citizens' lives. Others also argue that focusing too much on segregated dimensions or disciplines can make cities unable to reach their sustainability goals [10,13,20,21]. Bolívar [22] argues that the current practice of working in silos should be broken down with greater institutional integration, and governments should ensure that smart city efforts are coordinated rather than isolated. Many other studies also claim that the practical meaning of a smart city should necessarily emerge out of an interactive process of social dialogue and reflection, which demands that systems of governance guide and steer these collective discussions towards a satisfactory level of consensus [2–4,23–25]. They argue that the governance of sustainable smart cities should be interactive, implying that its practical bearing cannot be established independent of the needs, interests, values, and aspirations of its citizens [14]. Such interactions not only suggest the instrumental concept that societal input can facilitate progress towards smartness but also the deeper notion that the objectives of the smart cities themselves must be collectively defined, refined, and re-defined. Due to the rise of the WWW (World Wide Web), people's knowledge and information sources have increased, and their willingness and ability to participate and be an integral part of the decision-making process has become strengthened. The government, as the only institution with a general mandate, should be accountable to the general population and promote the public good through a democratic process [3]. In this regard, citizens need their governments who, in turn, need the intelligence and the cooperation of their citizens to function well. Correspondingly, as Meijer and Bolívar [3] (p.3) argue, "solving societal problems is not merely a question of developing good policies but much more a managerial question of organizing strong collaboration between government and other stakeholders", including citizens.

However, as the populations in cities increase, governments alone can no longer ensure a high quality of life for their city residents [3,9,22,26]. The new trends of privatization, outsourcing, and national fiscal retrenchment have also weakened governmental agency [9,27]. Therefore, the more effort that a society has put into developing more sustainable smart cities, the more clearly it has started to comprehend the full complexity of that task [11,22,28]. New systems of governance that are capable of putting society on a more sustainable path are clearly needed [2,3,18,26,29,30]. This demand for new and innovative forms of governance have yielded the term 'smart governance' [31], which many researchers agree is the core of smart cities [18].

While many scholars advocate using smart governance to overcome many challenges of our modern society, the question of which governance model or process is best (if any) is a question that remains under lively debate in both research and empirical practice. Building on earlier work [20,32–36] and aiming at understanding governance in different contexts, this paper empirically and conceptually investigates the governance in smart city initiatives. This article scrutinizes the effectiveness or usefulness of governance perspective in complex context of the smart city but developing and proposing an effective governance model is beyond the scope of our study. The methodological approach of this study aims at making sense of the many components that contribute towards the overall functionality of the governance system in smart city initiatives. Through a review and analysis of relevant academic articles from the smart city and governance literature combined with empirical support, this paper provides a critical foundation for analyzing the governance of smart city initiatives. The findings and discussion should encourage practitioners, researchers, and planners to consider the context in which the smart city occurs, how the governance system is structured and organized, and the way in which those structures interact with and contribute to the system's overall functionality.

The remainder of the paper begins with the theory of governance—the emergence of the governance approach—to distinguish it from the concepts of government and governing. We then discuss the potential challenges that can undermine the legitimacy and effectiveness of governance. With this perspective, the paper will review and analyze the relevant literature, understanding how different

scholars understand, investigate and/or deal with the governance of smart city. Providing insights from three Norwegian cities that are working to become smart allows us to reflect on their governance within this context, describing the complex reality of governing smart city projects. Finally, after a brief exploration of the dilemmas challenging governance efficiency, we reflect on the theory and practice of governance in a smart city and outline an agenda for further research.

## 2. Emergence of the Governance Approach

According to Jessop [37], the first uses of 'governance' occurred in the 14th century and denoted the action or manner of governing, guiding, or steering. However, in the last two decades, a surge in reverence for governance has been prompted by a persistent critique of the traditional hierarchical governing model, i.e., the government, for being undemocratic, expensive, and inefficient [38]. The advocacy of governance builds on the assertion that the formation and use of strategic alliances, partnerships, and interorganizational networks, as more flexible and proactive governing models, could compensate for the failure of the bureaucratic control of the government. In the 1980s, a neoliberal form of governing, with less state involvement and more market involvement, was proposed as a solution to urban problems [39]. The invisible hand of the market was suggested to not only ensure an optimal allocation of private goods, but also to regulate the production of public goods more efficiently [37,38,40,41]. In a neoliberal setting, the implementation of competitive regimes of resource allocation justifies the public goodness of privatization, lean government, and deregulation [42]. Later, the neoliberal form of governance manifested itself in the theory of new public management (NPM), which contains two primary components: managerialism and new institutional economics [38]. The former entails bringing private and public sectors together, while the latter introduced incentive structures, such as market competition, into public service provisions [38]. The parallel financial cutbacks also forced cities to rely more and more on private and local sources of revenue [43]. In this respect, Cocchia [28] connects the availability of European financial resources earmarked for smart city projects with the current allocation policies of cities hit by the economic crisis and international competition. The argument of collaboration, particularly between public and private partners, has created a 'networked' governance approach, in which "governments are no longer called upon to govern, command, and control but to steer" societies [44] (p.46). Indeed, according to Sørensen and Torfing [45] (p.236), the governance network has become a stable articulation of mutually dependent, but operationally autonomous actors who interact within a self-institutionalized framework of rules, norms, shared knowledge, and social imaginaries.

We must distinguish the concept of "governance" from "governing" or "government", which are not interchangeable [46]. Governing refers to the social activities that make a "purposeful effort to guide, steer, control, or manage (sectors or facets of) societies" [47] (p.2). It should be clarified that governing today embodies 'government' and 'governance' as two related and intertwined processes [48]. Government, as a sphere of authority combining legal, financial, political, formal, and institutional processes, operates at the level of the nation state and its subdivisions to maintain the public order and facilitate collective action. "Government" is based on centralization and control, whose relationships with other units of the policy network are asymmetric [38]. Conversely, governance operates under shared goals, as a sphere of public debate, partnership, interaction, dialogue, and potential conflict among all stakeholders [48]. Therefore, governance refers to "a shift from state sponsorship of economic and social programs and projects to the delivery of these through partnership arrangements which usually involve both governmental and non-governmental organizations" [49] (p.41). While governance refers to activities based on shared goals that may or may not derive from legal or formally prescribed responsibilities [50] (p.4), government suggests activities that are backed by a formal authority [51] (p.4). According to this argument, governance is not the complete opposite of government. It is, however, a more encompassing concept, which brings both governmental and non-governmental institutions together. Furthermore, the outputs of governance are not necessarily different from those of government. Rather, the processes are different [52].

In this respect, the governance of a smart city requires that different stakeholders, including citizens, should take part in the planning and decision-making processes, share control over the development of initiatives, collaboratively address problems, and set priorities to build commitment and ownership of the final planning outcomes [29,52–55]. In addition, the effective role of governments in the governance of smart cities is essential, as here, governments can play the role of coordinator, funder, and regulator by bringing different interests and stakeholders together, providing funding for infrastructure and demonstrator projects, and ensuring that common standards and regulations are in place [22].

Sørensen and Torfing [45] argue that even though traditional forms of top-down government (should) remain in place, governance increasingly proceeds on the basis of interdependency, trust, and jointly developed rules, norms, and discourses. However, this does not mean that a governance network is formed by agreed norms, procedures, or constitutions to assess legitimacy or efficiency [56] (p.341). Indeed, the formulation of a framework for rules, norms, values, and ideas in governance networks is usually precarious and incomplete. The next section discusses some aspects of this precariousness, i.e., the challenges of a governance network.

*Challenges of Governance as A Mode of Governing*

As discussed, the reliance on governance networks is often justified by the need to enhance the effectiveness of public governance. However, the governance concept has already faced many criticisms/questions regarding its efficiency, democracy, accountability, and legitimacy that no theorists can defend perfectly [57]. Sørensen and Torfing [45] argue that the lack of accountability and the privileging of strong and resourceful elites is a ubiquitous danger in a networked policy of governance. In order to fulfill responsibilities, there must be an authority—a legitimated power [58]. However, in governance networks, the centers of power are multiple and fluid. Since the necessary powers are not possessed in the governance systems, they have to be granted, either with the help of knowledge, skills, and experience or through informal networks (who we know and who knows us) [58]. Therefore, power can be gained through a potential opportunity that an actor realizes in order to turn the table on him/herself [59].

In governance networks, there is no single actor who has enough steering capacity to determine the strategic actions of the other actors, but each actor can be seen as a potential leverage point within the political stream [60] and can actively promote policy options or solutions [57,61–64]. Actors' influences on each other's actions and policy outcomes make the processes of bargaining, coalition formation, and conflict mediation imperative [29]. Each actor, based on his or her position and subsequent legal rules, has specific responsibilities and formal competences required to intervene in the planning and decision-making processes, which determine the possible actions or role (s)he can take. However, the action or role of an actor is a function (s)he fulfills within the process and through interactions with other actors, which are not only affected by the formal position (s)he has but by the dynamics of that specific process. This means that, in different processes or even in the same process, the same role (e.g., being an initiator, leader, enabler, regulator, promoter, ally, mediator, opposer, or gatekeeper) can be played by politicians, planners, developers, or bureaucrats [65]. In these processes, many actors may be forced or convinced to change their attitudes and roles, thus setting new goals, which may differ from their original and real interests. For instance, politicians and public managers, whose responsibility is to defend public interests, may seek to exploit their privileged position to pursue particularistic interests [66]. Based on actors' new goals, new networks will be formed, and actors may play new roles. Such loops can be repeated again and again until a particular condition is satisfied [66]. The same kind of interactions may result in different outcomes if they are formed at different times. Therefore, the temporal aspect of these interactions (i.e., when these interactions take place) is important to be considered [66]. Not only do actors change with time, but society, technology, people's values, beliefs, cultures, and positions also change. In addition, there are many different influential external factors that change over time and increase the complexity of governance networks, such as swings

in the national mood/agenda or new election results and subsequent new regulations, new systems and new environmental conditions, etc. [64]. Therefore, the dynamic interactions of actors' interests, roles, and power, which fluctuate continuously, can affect the outcome or sustainability of smart city initiatives [57].

Benditt [67] considers 'time' to be an effective factor in interest theory and believed that one's interest has two dimensions: (1) what sorts of things are in one's interest and (2) what is in one's interest at a particular time. He points to the fact that sometimes it is not in a person's best interest to pursue one of his/her interests. In addition, 'an actor's interest' and 'the best thing to do' are different notions [67]. Therefore, actors' actions do not necessarily reflect their real interests at a given time. Furthermore, an actor's chances to fulfill his/her interests are likely to depend on other influential physiological, subjective, and behavioral attributes, such as available resources and power. Therefore, some policies are in one's best interest, as long as one has specific resources or power. The same policy can be against one's interest during another period. The discussions of this section can help develop a critical understanding of the values and ideas that underpin the smart city concept and how they are translated into practice, as follows in the next section.

Potential Challenges for the Governance of Smart Cities

Torfing and Sørensen [68] argue that governance theory can give 'unfettered' power to specific groups of people to do almost whatever they want. In the smart city context, Meijer, Gil-Garcia [30] share the same belief, arguing that the urban governance of smart cities has clearly been shaped and steered by large and influential commercial players in the hardware, software, and infrastructure sectors, such as IBM, General Electric, Cisco Systems, Hitachi, and Siemens. Per Guarneros-Meza and Geddes [69] and Blanco [70], smart city power is increasingly concentrated in the hands of these political and business elites, who skillfully promote a story of urban problems in ways that position their own services as the only solution. Hollands [71] notes the fact that governance plays into the hands of special interests, who benefit from an increased emphasis on efficiency savings, privatization, and the promise of a high-tech future resulting from the general trajectory of 'neo-liberal urban utopias' and the new worldwide politics of austerity. While these authors are concerned that the governance of smart cities neglects the need for political (not only technological) answers to public and common interests, many scholars highlight the creation of public value as the main outcome of smart city governance [30].

On one hand, Grossi and Pianezzi [72] argue that global competition, privatization, partnership with private sectors, and neoliberal ideology in general lead to business-driven development and technological solutions, which might result in a prioritization of business goals over social, political, and environmental ones. Therefore, it is difficult to estimate to what extent existing smart city planning or decision-making processes keep up with a city's original expectations and ideals, such as enhanced citizen participation, sustainability, etc.

On the other hand, the fact that people do not appraise things similarly challenges the achievement of common goals [73,74]. While the aggregation of individual preferences is itself a highly complex matter due to the diversity of conflicting interests, there is considerable dispute and uncertainty regarding whether there is any group interest or public value other than the sum of individual preferences. Baron [75] argues that collective interests/goals are not necessarily agreed-upon values, and vice versa. On the contrary, a private or particular interest can accrue among one or a few actors at the expense of benefits to others. Achieving a result that satisfies everyone at once is sometimes impossible. This difficulty is exacerbated when different actors' interests and values are involved in a pluralist and complex context of planning and decision-making, such as in a smart city. Each of the various actors has his or her own interests and perceptions of the issue. Therefore, their desired solutions may be contradictory. Actors also have different perceptions of other actors' interests in the network, on the basis of which they select strategies [73]. The interactions between different actors' strategies shape the outcome of the planning and decision-making processes [76]. Therefore, it is

difficult to estimate to what extent the existing smart city planning or decision-making processes keep up with the city's original expectations and ideals, such as smartness or the capability to provide infrastructures and services that improve the lives of its citizens (one of the common goals that many cities are trying to achieve) [77].

While the EU considerably influences the development of smart cities in Europe, Haarstad [78] argues that it cannot impose a particular ideology on cities. On the contrary, actors across sectors are able to co-define their own values, agendas, and strategies, thereby lending efficiency and legitimacy to locally initiated projects. These two sides of the argument make an assessment of governance crucial in order to maximize its merits and minimize its problems. In order to avoid replicating bad practice while trying to benchmark good practice, this paper first disentangles and analyzes the relevant literature on the governance of/in smart cities. The aim of this study is not to provide an exhaustive or comprehensive review but rather to identify what has been done and should be done.

## 3. Governance in the Smart City Context

According to Meijer and Bolívar [3] (p.398), "the smartness of a city refers to its ability to attract human capital and to mobilize this human capital in collaborations between the various (organized and individual) actors through the use of information and communication technologies (ICT)". Many scholars agree that technology alone will not make a city smarter [2,4,55,79,80]. Indeed, as Meijer and Bolívar [81] and Nam and Pardo [4], [55] argue, we should determine the underlying political paradigms, institutional structures, and interactive processes within which smart city initiatives are undertaken (i.e., urban governance). In this respect, Taylor Buck and While [9] also show how a poor understanding of urban governance and an underestimation the importance of 'soft' human infrastructures has undermined the implementation of smart city initiatives. According to the literature review, there is a consensus that smart governance plays an essential role in improving government systems, involving various stakeholders, offering equal citizen engagement opportunities, and transparent information for relevant sharing mechanisms [24]. However, different scholars offer different interpretations of smart governance. For some authors, smart governance is about the integration of Information and Communication Technologies (ICTs) in the internal administrative operations of governments to achieve governability, efficiency, openness, transparency, accountability, inclusion, and equality [2,3,53,82–85], a so-called e-government. For others, smart governance refers to the governing process of a city that promotes itself as smart [15,16,23,71,78,86]. The latter group focuses on the negotiated involvement of multiple public and private stakeholders operating at different scales. Here, a governance model can be seen as the pattern or structure that emerges in a socio-political system as a 'common' result or outcome of the interacting intervention efforts of all involved actors [18,54]. As Meijer and Bolívar [3] argue, for ICT-enabled governance, the focus is mainly on strengthening the legitimacy and content of government actions, whereas for the governing of smart cities (the latter approach), the authors focus on the process of governance. The publications that frame smart governance as a technical or managerial issue envision a futuristic city that offers a high quality of life for everyone, which requires legitimate governance to actualize. However, according to Meijer and Bolívar [81] and Dameri and Benevolo [87], there is a lack of attention on the politics of a smart city: "who and what is driving smart city, and who stands to gain and lose in the race towards such an urban future?" [71] (p.62). Hollands [71] argues that the question of legitimate governance should be subordinate to the legitimacy of a smart city, for which such a corporately envisaged utopian future is assumed to be what all people want and is in everyone's interest. Correspondingly, as Meijer and Bolívar [2] note, enhanced citizen participation as a source of government legitimacy has so far been analyzed as a desirable outcome of a smart city, instead of being an issue of political struggle. The reason for this focus is that smart cities are assumed to be an issue of 'puzzling' rather than 'powering' [2] (p.403). Accordingly, before investigating smart governance, Castelnovo and Misuraca [1] recommend us to reassess whether the planned future full of ICTs that we are moving towards is the right direction to go. This requires a systematic understanding of the nature of the governance models used in smart city initiatives [30]. According to Alawadhi

and Scholl [29], few studies thus far have inquired into the emerging governance models in smart city initiatives. In response, Alawadhi and Scholl conducted one of the first (comparative) empirical studies on smart governance initiatives to explore the characteristics of governance models in smart city initiatives in three cities (Seattle/USA, Munich/Germany, and Turin/Italy). They looked at the structure of governance, the roles and responsibilities of the officials involved, and the decision-making process regarding prioritization, information sharing, and conflict resolution. Their conclusion was that "all governance models seem to be capable of effectively supporting the objectives of the given smart initiative" [29] (p.2962). However, their study neither reflects on the politics of smart city governance nor represents a real image of the complex, unpredictable, and uncertain context of a smart city. They instead adopt a more static perspective, which simply describes the presence of particular modes or instruments of governing without exploring the relationship between governance structures and functions (i.e., the interactions of power, interests, and actions) in a more dynamic and interactive manner.

This paper fills the gap by exploring the causal relationship between governance structures (the way actors exist in a network in relation to each other across sectors and levels) and processes (how actors' interests, roles, and powers influence the outcomes). This is in line with the argument of Cowell and Owens [88] that the choice of governing mode will never be a totally open or value-free activity. Smart governance is widely and often uncritically identified as a "good thing", but we need to know why, when, and to what extent it can affect the ends for which it is designed, clarifying the question of 'governance for what and for whom?'.

## 4. Methodology

### 4.1. Research Justification

This paper contributes to depicting the reality and complexity of governance practices in different smart city initiation processes, thereby addressing which actors, processes, and relational mechanisms at different levels of governance can influence the development of smart cites in Norway. The main questions are as follows:

How have governance networks formed and influenced the smart city initiatives in Trondheim, Bodø, and Bergen?

Which collective actors are involved? what are their respective goals/interests, roles and power in developing the smart city initiatives?

Many scholars argue that governance functions differently in response to divergent conditions of localities, societal and technological context, institutional design, organizational and managerial culture, and political struggles [4,30,45,53]. There is no uniform governance model or practice for smart city initiatives [1]. For example, a long history of autocratic governance, conflict, or distrust may imbed a distinctly non-collaborative practice of governance, despite a will to collaborate. Therefore, benchmarking a governance structure or function should respond to the cities' particular histories, needs, and practices, without being subjugated by their contexts [89].

Governance models sometimes need immediate, radical, and uncompromising modifications, for example in response to a cutback in government funding or a crisis of confidence in leadership. In order to modify a governance model, an understanding is needed of what the pre-existing governance model lacks and what a particular modification might accomplish. The investigation and evaluation of governance thus requires careful and periodic reviews, including identifying how governance functions in practice and how it might be modified to work better [90].

Norway has persistently and narrowly followed sustainable development goals, recently through a smart city approach in which digital technologies are enablers of smart solutions. Smart technologies are not considered to be a goal but rather a means to achieving sustainability. Ødegaard [91] argued that the implementation of smart technologies in Norway has strong financial motives but yields environmental and social profits, thus improving the quality of life for citizens while reducing the

carbon footprint. Although many Norwegian cities have given themselves a smart label, no Norwegian region has come as far as its European counterparts when it comes to smart city initiatives. Most smart city strategies remain in the planning stages and are still very fragmented [15,78,91]. According to Ødegaard [91], Norwegian smart cities are somewhat arbitrary and "assembled piecemeal, integrated awkwardly into the existing configuration of urban governance and the built environment" [92].

This paper chooses three Norwegian cities, Trondheim, Bergen, and Bodø, as typical cases [93], to probe the causal mechanisms of governance processes in the development and initiation of their smart city strategies. The consideration of such ongoing processes gives an extra scope to this research, since this analysis can reveal some of the existing tensions/deficiencies and provide new lessons and guidelines for forthcoming actions.

### 4.2. Research Design

The scope of the governance study in this paper encompasses an analysis of empirical and conceptual content. However, this paper acknowledges that "the study of governance is complicated by its broad scope and defining elements, the nature of its configuration, the political interests and activities that shape it and exercise influence, and the formal and informal rules and authority that characterize the execution of public policies" [94] (p.3). According to Heinrich and Lynn Jr [94], there is a methodological challenge to the study of governance due to its inherently political nature, involving bargaining and compromise, as well as ambiguity and uncertainty. Smart city initiatives are also complex and context-specific [95]. According to George, Bennett [96] and Heinrich and Lynn Jr [94], qualitative empirical case studies are the most appropriate approach for investigating the subtle complexities of governance. Research that employs theory, empirical methods, and systematically obtained data and observations is likely to be the primary source of fundamental and durable knowledge that transcends particular times, places and contexts [94]. In addition, this approach enables the exploration of actors and processes and the mechanisms that exist between them across different levels. In order to better explore the causal mechanisms of governance elements (actors, interests, roles, and power), this paper performs a pattern-matching investigation, in which the evidence in our case is judged according to what has been stipulated in Section 2 [93]. The results may show that the causal mechanisms are different than those that had been previously stipulated (e.g., in the study of Alawadhi and Scholl [29]) or confirm them. In this regard, our selected case studies are typical case studies of smart city governance, which represent complex, uncertain, and multi-layered systems.

In addition, this paper relies on community-based participatory research (CBPR) [97,98], in which researchers are a part of the Smart Sustainable Cities research group at the Norwegian University of Science and Technology (NTNU) and participate directly or indirectly in a number of projects and initiatives. CBPR is a form of collective action that a community, in which researchers are a part, can undertake as a key to the community's survival and empowerment of its continued effectiveness in encouraging social and political change [98]. The aim of CBPR is to increase the knowledge and understanding of a given phenomenon and to integrate the knowledge gained with interventions for policy or social change benefiting the community members. Therefore, researchers, as members of the social world that they study, can contribute and share their expertise in the Norwegian planning and decision-making processes for smart city development. Through active participation, observations, and informal discussions with different partners, researchers can reflect, refine, and disseminate their governance approaches and potentially affect the behavior of individuals or systems [95,98]. This approach is called reflexivity [99] (p.98) and involves introspection into the role of subjectivity and researchers' reflections upon their own values in the research process. This is a continuous process of recognizing, examining, and understanding how a researcher's "social background, location, and assumptions affect their research practice" [100] (p.17). On the other hand, this paper acknowledges that it is never possible to fully understand the motivations and purposes that compel researchers towards a particular understanding. Therefore, this research remains open to critical evaluation and reconsideration and is flexible to the ongoing dynamics of individual and group development and

change [94]. By articulating the dimensions of a community-based research issue, this paper aims to make a positive contribution to bringing researchers and practitioners a step closer to each other at a time when more cooperation can serve both sides.

In our case study research, we have chosen convenience sampling as a type of nonprobability or nonrandom sampling due to the researchers' involvement and subsequent accessibility and networks. This type of sampling is used because our cases are typical/homogeneous, and the aim is not to create generalizations pertaining to the entire smart city context [101]. The selected cases are related to ongoing projects or collaborations, and our sampling seeks to cover different types of environments for governance structures. We also tried to choose cases that represent an appropriate size and involve different types of stakeholders, based on a quadruple helix model [102,103] involving public actors at different levels (international, national, and regional–local), the private sector, universities (considering their multilayered spatial ties [104]), and civil society/citizens. The synchronization and coordination of planning activities between national, regional, and municipal levels suggest that all sectors function across different levels of society. Even though we acknowledge that sectors are organized or institutionalized in a specific manner and consist of many interrelated, interdependent, and autonomous individuals at different levels of society ((super)national, regional and local), our focus is only on the collective/organizational dimension of actors at each societal level. In this regard, we map out actors based on their roles and positions in the project at different levels, e.g., municipalities at local level and EU at international level. However, it is difficult to structure the university at one specific societal level because universities by their nature are bi-focal or multi-focal organizations and operate between local, national, and global poles [104]. In a smart city context, it is thus unclear at what level universities act, since universities simultaneously play different roles (teaching, research, innovation, and transfer). Accordingly, the level of university interests/goals is also multi-layered.

In addition to the position of each actor in a network and in relation to each other (the reality of the actors' (formal) interactions), smart city initiatives are influenced by the interactions of actors' interests/goals, their roles, and the uneven distribution of power. In this regard, the formal and informal powers of actors, together with their roles as regulators, enablers, initiators, mediators, allies, opposers, or gatekeepers, are identified. The features in the tables in Section 5 (see Table 1, Table 2, and Table 3) were identified by at least one researcher familiar with the case and discussed and approved in the discussion group.

## 5. Cases

The following section describes the three selected cases in more detail and discusses their governance structures, following the methodology and study design outlined in Section 4.

### 5.1. Trondheim Case

Trondheim is developing different smart city strategies and projects at different organizational levels. It is also a partner in the +CityxChange smart city project, funded by the EU Horizon 2020 research and innovation program in the topic 'Smart Cities and Communities'. The Norwegian University of Science and Technology (NTNU) is coordinating the project with the lighthouse cities of Trondheim and Limerick, five follower cities in Europe, and 24 other private and public partners [35]. The main goal of this project is to co-create positive energy blocks and districts (PEB/PED) through integrated planning, common energy markets, citizen participation, regulatory sandboxes, and business models. This paper studies the +CityxChange pilot project in Trondheim.

This project grew from the collaboration of a small number of core players at Norwegian University of Science and Technology (NTNU) and Trondheim Municipality to include additional partners as needed for the EU's initiative. Therefore, the university and municipality were the initiators and promoters of the project. The most important objective of the initiation process was to align and anchor smart city development in the form of creating PEDs with the city's processes and strategies.

In this project, the EU through H2020, at the super national level, has played the role of regulator and financial enabler, to help secure Europe's global competitiveness and sustainability based on its three pillars. Due to H2020's role and position in this project, the EU's power is mainly formal and legal. However, it can also informally influence the actions of other actors, as well as the direction and outcomes of the process.

Due to its common political interests, the Norwegian national government aims at improving its international competitiveness and following all three pillars of sustainability. The regional government also seeks a regional transition in its energy markets due to its economic and environmental interests. The national and regional governments are political allies that have a more informal influence than formal power/authority in this project.

At the local level, the municipality and university are the main actors. The municipality is the main initiator and leader and aims at industrial competitiveness and environmental sustainability, sustainable energy transitions, and improving the quality of life in the city, due to political, economic, environmental, and social interests. The United Nations Economic Commission for Europe (UNECE) and the Municipality of Trondheim have agreed to establish a Geneva United Nations Charter Centre of Excellence to advance sustainable urban development and improve the quality of life through training, studies, awareness raising, and concrete projects on the ground [105]. In addition, Trondheim will work closely with the United for Smart Sustainable Cities (U4SSC) initiative, which was developed by UNECE, ITU, and the United Nations (UN) Habitat in collaboration with 13 other UN agencies. The scope of this initiative is to evaluate the performance and potential for smart sustainable development in cities and communities and connect local needs to a global knowledge hub, solution providers, and funding opportunities [105]. In addition to the delegated authority/formal power across levels, the municipality can informally influence the project due to its political interconnections.

NTNU as a multi-sectoral organization, which serves different users across different levels, pursues different aims, and plays different roles. In accordance with its international competitiveness goal and political–economic interests, NTNU participates in EU research, contributes to solving climate and energy issues, and thus has formal power. Due to having the same types of interests, NTNU pursues national competitiveness and seeks to be a mediator between different cities, thereby informally influencing the process. At the local level, NTNU is the leader and enabler that realizes the EU's financial opportunities and uses both formal and informal networks/power to facilitate better collaboration with local and regional partners. Accordingly, its interests are political and economic, as well as environmental and social.

Partners in industry are also allies and the enablers, providing the right technology, and aiming at innovation, economic and energy competitiveness, and technical knowledge development. Their main interest is economic, with both forms of formal and informal power. Both the university and municipality have attempted to involve organizations and people in different projects, mainly on a trial basis to determine how they can fit their inputs within the project. However, people's engagement has been more informal than formally institutionalized. Table 1 represents the governance system across public, private and civil society sectors (shown by different colors) in the Trondheim case. The role, interest, and power of university, as a multi-sectoral organization, are addressed at the different levels of i. international, ii. national, and iii. regional–local.

**Table 1.** Governance in the development of a smart city in Trondheim.

| | | | Goal/Interest | Role | Power |
|---|---|---|---|---|---|
| **Public Sector** | | | | | |
| | **Super-national (EU Horizon 2020)** | | Industrial competitiveness and sustainable energy/three pillars of Sustainability | Regulator, Financial Enabler | Formal–informal |
| | **National** | | International competitiveness/three pillars of Sustainability + political interest | Ally (financial and political) | Informal |
| | **Regional** | | Regional transition in energy market/Economic and Environmental interests | Ally | Formal |
| **Local** | **Municipality** | | Industrial competitiveness and environmentally sustainability/Political, economic, environmental, and social interests | Main initiator, leader | Formal–informal |
| | **University** | **International** | i. Participation in EU research/solving climate and energy issues/International competitiveness/Political and economic interests | Initiator | Formal |
| | | **National** | ii. National competitiveness/Political and economic interests | Ally, mediator (connecting cities) | Informal |
| | | **Regional–Local** | iii. Better collaboration with local-regional actors/Political, economic, environmental, and social interests/development and application of research/contributing to local solutions | Leader, enabler (realizing the finance) | Informal–Formal |
| **Private Sector: Industry partners** | | | Innovation, economic and energy competitiveness. Knowledge development about their technology and service/Economic interest | Ally, enabler (providing technology) | Formal–informal |
| **Civil Society Sector: Citizens** | | | Accessible, efficient city services, participation in energy transition | Passive, (recipient, user, consumer [106]) | Informal |

## 5.2. Bergen Case

There is no comprehensive smart city strategy in Bergen yet [91]. Several smart initiatives have been undertaken with a focus on innovation and entrepreneurship. For example, SmartCityBergen is a public–private innovation project that aims at creating a hub, i.e., a network of people/ideas/competences/companies and a meeting place for the coordination and discovery of projects concerning digitization and Internet of Things (IoT) [107]. Another smart city initiative that this paper focuses on is the Zero Village Bergen (ZVB) project, a pilot project at the Research Centre on Zero Emission Neighborhoods (ZEN) at the Norwegian University of Science and Technology (NTNU) in Trondheim, started in 2009. The ZEN center serves as an innovation hub for co-creation between different stakeholders across sectors, functioning as a lighthouse to develop solutions in real-life contexts to support the development and dissemination of ZEN-related knowledge [33]. Correspondingly, the main focus of the Bergen Smart City is on buildings, including commercial buildings, public buildings, lighting in industry, and street lighting, as well as upgrading the grid and making households 'energy smarter'.

In Bergen, the private sector is the main initiator, leader, and provider of the technology and exerts a large amount of informal influence, as well as some formal power (due to its cognitive and economic resources). The municipality is the main ally of the private sector, who has economic and political interests, with a great amount of informal influence. The private sector has had more formalized communication with the property division of the municipality in order to choose sites for development. Due to its economic interests, the private sector has shown interest in agricultural land that has not been regulated for housing purposes. The selection of this site has created conflict with several public agencies, among which the county governor has been the strongest opposer. For many years, the realization of the ZVB has stagnated due to existing conflicts. In 2019, the area plan for ZVB was finally approved by the Ministry of Local Government and Administration, which plays the role of mediator between the local and regional governments and the private sector. In addition, due to the government's responsibility in enacting the relevant laws and regulation (and thus formal power/authority), the government has played the role of regulator, whose interests focus on international competitiveness from different social, economic, environmental, and political perspectives.

In the current stage of the planning process of ZVB, there has been no direct community involvement beyond that of elected politicians. Thus, the people's involvement and power are not (yet) formally recognized [33]. Table 2 presents the governance network across public, private and civil society sectors (shown by different colors) for Bergen city.

This section is not mandatory but may be added if there are patents resulting from the work reported in this manuscript.

## 5.3. Bodø Case

Bodø has strong smart city ambitions and is also one of the pilot projects in ZEN, where a former airport is planned to be moved and its area to be turned into a new city district. Therefore, expanding the existing city centre is a parallel strategy that offers the possibility to develop a dense and mixed-used urban neighborhood that is environmentally friendly and citizen-centered and can contribute to business and urban development.

While the re-location of the Bodø airport was on the agenda, one of the employees of the Confederation of Norwegian Enterprise (NHO) in Bodø was involved in writing an application for the EU's research program, Horizon 2020, which prompted the initiation of the Bodø smart city project. In the beginning, specific individuals from the private sector, who did not have formal positions or power, initiated the smart city idea and discussed it with the municipality. Their interest was mainly economic and political. Today, these individuals have been appointed to more formal and critical positions due to their former roles as initiators, there granting them more legal and political resources to be able to work as the leaders and enablers of the project. These stakeholders were successful in shifting their informal influence to a more formal type of influence, gaining authority by assuming formal positions

at the public sector. In addition, they effectively managed to convince the government at all the levels to become their own allies. Today, the smart city has become the encompassing strategic vision in Bodø, where the municipality involves a broad spectrum of actors, facilitating the collaboration between them and aiming at co-creation of solutions to future challenges. However, concrete progress remains forthcoming, and more research is needed to assess whether this comprehensive outlook will persist. Similar to the Bergen case, people in Bodø are considered to be the center of the project, even though their power has not yet been legitimized. Table 3 presents the governance network across public, private and civil society sectors (shown by different colors) for Bodø city.

**Table 2.** Governance in the development of a smart city in Bergen.

| Public Sector | | Goal/Interest | Role | Power (Formal–Informal) |
|---|---|---|---|---|
| National | Government | International competitiveness/three pillars of Sustainability and political interests | Regulator and Mediator | Formal |
| | Research center (ZEN) | Environmental | Initiator and promoter | Informal |
| Regional government | | Follow regulations and social interest | Regulator, Opposer, and gatekeeper | Formal |
| Local government | | Industrial competitiveness and environmental sustainability/Political, economic, environmental, and social interests | Enabler and Ally | Informal and Formal |
| Private Sector: Industry partners | | Technology provider, economic interests | Promoter and Leader | Informal and Formal |
| Civil Society Sector: Citizens | | Accessible, efficient, and cheaper city services/Economic interests | Passive | Informal |

**Table 3.** Governance in the development of a smart city in Bodø.

| Public Sector: | | Goal/Interest | Role | Power (Formal–Informal) |
|---|---|---|---|---|
| National | Government | International competitiveness/three pillars of Sustainability and political interest | Regulator | Formal |
| | Research center (ZEN) | Environmental | Promoter and ally | Formal |
| Regional government | | Regional transition in efficient transportation and energy market/Economic and Environmental interests | Ally | Formal |
| Local government | | Industrial competitiveness and environmental sustainability/Political, economic, environmental, and social interests | Enabler and ally, later leader | Informal–Formal |
| Private Sector: Industrial partner | | Innovation in economic competitiveness and knowledge development about their technology and services/Economic and Political interests | Enabler and ally | Informal |
| Civil Society Sector: | | | | |
| Specific individuals (coalition between two people from the private sector and one from the municipality) | | It is difficult to identify the individual goals/Political and Economic interests | Initiator, promoter, and enabler | Informal–formal |
| Citizens as a collective actor | | Accessible, efficient, and cheaper city services/Economic interests | Passive | Formal |

## 6. Findings

The findings show that the initiation of these smart city projects in Norway is primarily influenced by the informal interactions of outside actors or in the shadow of actors' formal positions and relationships. Such initiatives might never enter the political agenda or policy domain without actors'

informal interactions. The smart city initiative in Trondheim emerged from an informal network of university and municipal administration, in Bergen, from an informal network between the private sector and municipal politicians, and in Bodø, from an informal network between individuals in the private sector and the municipality.

The actors apparently use their interpersonal connections to integrate their expertise or influence into the definition of the problems or the shapes of the problems (e.g., in the shadow of sustainability or smartness) so that they can drive processes of change or benefit from their results or policies. In addition, they can push municipalities (mainly politicians) to pass certain types of policies and favor some policies over others, thereby asserting their informal power. Such informality may compel actors to engage in tactical ploys of direct action, politics, and power-plays [108]. Actors may be motivated by factors like a sincere concern for a particular problem, sustainability principles, self-serving benefits, the potential to integrate their own policy values, or gaining personal satisfaction from simply participating in the process. Thus, smart city research should focus on whose interests are being served or pushed by the organized and coordinated activities in the initiation phases and whether they align with societal or citizen-oriented benefits. At the national and regional levels, where the governance network is more formal and authorized, it is easier to realize actors' goals. However, at the local level, where the governance system is more informal and actors' actions are largely based on their informal power, the identification of their real goals challenging. On the other hand, the fragmentation of responsibilities at the local level has made smart city planning more complex, ambiguous, and uncertain, which challenges for transparency, accountability, and legitimacy, particularly in the initial processes. The informality of governance networks recalls the significance of the government's role and interventions in establishing more formal/legitimate platforms for collaboration (acting as a coordinator), changing innovation and incentive structures (enabler), funding infrastructure and demonstrator projects (acting as a funder), and making sure that common standards and regulations are in place (acting as a regulator). It is impossible to fully 'depoliticize' smart planning decisions, in which actors' roles/logic are linked not to their formal positions, but to the informal dynamics of the processes they are taking part in [108].

With reference to the Trondheim case (which is now in the implementation phase), we have witnessed a shift of governance from an informal structure to a more formal one. Even though all the partners' roles, goals, and power are clear and transparent enough for everyone to follow and understand, the governance functionality in the area is not completely stable and continues to evolve. The actors are not stable but varying constantly, i.e., they constantly enter and exit the governance network. Not all actors involved in the initiation phase or writing the proposal remain a part of the project today. The same is true for the actors' interests, goals, preferences, resources, and roles, which can exacerbate existing conflicts and cut off the mainstream of preplanned actions or pre-determined milestones and contract deliverables. While this project has a time limitation, the actors that have recently joined the process require more time to learn about the whole project, the rules of the game, the interconnections between different parts, and other actors' roles and strategies. In addition, the proposal of the project was written when knowledge about different social, political, and economic aspects were naturally uncertain due to the long-term characteristics of strategic planning. Correspondingly, enacting strategies for change or adaptation requires extra time and resources. The time-consuming and resource-intensive involvement of different actors in the planning and decision processes conflicts with the time-limited constraints of EU smart projects. Thus, the efficiency of the project and the implementation of smart city initiation is challenged by the unpredictable nature of smart city development, as in many other projects. However, the evolving smart city structure and landscape also opens up new opportunities and will ideally help to bring different initiatives together to evolve the overall strategy

One additional note on these findings is that we examined large-scale initiatives. In these initiatives, there will almost always be a city involved as a formal actor. Smart cities are made up of many small features that are not all required to be directly connected to the municipality and, in

fact, often have informal and unbounded governance structures. These are important backbones for activities within cities, but once initiatives reach a certain scale or impact ambition, it is highly beneficial to align them with existing formal city governance structures (such an action may even be mandated). This also leaves many smaller-scale governance structures and functions that have not yet reached high visibility out of the present study's scope, but they will be interesting for future work.

## 7. Conclusions

Many scholars agree that governance, which is based on negotiation and collaborative rationality, is fundamental to the design and implementation of smart city initiatives. When it comes to enhancing the legitimacy of the regulatory standards issued by the European Union (EU), political decision makers increasingly turn to network-based forms of governance, building on the horizontal interactions between public and private policy actors. Even though the European Commission has branded its funding programs for creating smart cities with a dedicated governance focus, it is not clear how a smart city should govern itself so that the sum of its human development becomes more sustainable in the long term.

Since the choice of governance is not a totally open or value-free activity, this paper has tried to develop a critical understanding of the values and ideas that underpin the smart city concept and the way they are translated into practice. In order to make sense of the many components that contribute towards the overall functionality of the governance system in smart city initiatives, we have asked: How have governance networks formed and influenced the smart city initiatives in Trondheim, Bodø and Bergen? Which collective actors were involved? What were their respective goals/interests, roles and power in developing the smart city initiatives?

This paper has depicted the real complexity and challenges of governing smart city initiatives by using three cases and accumulating political knowledge. Our results can help academics, planners, and politicians better understand some of the factors that can be learned from the practical realities of governance politics in similar smart projects. This paper has discussed the potential of different approaches for project initiation and also highlights both the theoretical and methodical challenges in the study of governance. Due to a certain informality in the processes through which these cities are becoming smart, it is difficult to completely grasp the corresponding political sensitivities and behind-the-scenes negotiations and determine the real goals and interests of the involved actors. Therefore, the legitimacy and efficacy of smart city governance may be challenged by common pitfalls, such as informal/interpersonal networks, hidden relationships between actors, opaque leadership, a lack of accountability and control functions, uncertainty about access to critical resources, and a lack of transparency for interests and strategies.

We view this paper as a starting point for exploring the governance approach to smart city initiatives. It would be valuable to further explore the differences and similarities between our Norwegian cases and other national or international cases in future studies. There is certainly a need to undertake more detailed empirical studies to explore how governance functions in different contexts and phases of smart city planning, as well as to investigate governance's interrelationship with other modes, including systems of regulation, government control functions, and hierarchical coordination. In order to evaluate the efficiency, legitimacy, and sustainability of smart city projects and the effectiveness of the recent governance model in complex, unpredictable, and non-linear smart city decision processes, a more holistic theoretical framework is needed. Such a framework should go beyond legal frameworks, formal institutions, and processes, in which the political structures underpinning the (informal) functioning of governance in terms of different actors' roles, interests, resources, and power are usually investigated. An exclusive focus on smart city projects, without considering the political systems that form them, obscures the complexity and reality of planning and decision-making processes. However, smart cities are an application domain where governance issues are highly relevant due to their complexity on top of existing governance systems. In addition, it is extremely difficult to understand the network relations at each individual level (national, regional, or

local) without considering the evolution of the interconnections between levels over time. A holistic analysis of governance can allow non-political actors, including planners, to unravel the political planning and decision-making processes and better grasp their rationale. This type of analysis provides a better understanding of the national political mood/priorities, potential contradictory interests, strategic roles, and interactions between formal and informal networks in the ways actors shape, change, or develop the relevant processes.

**Author Contributions:** Conceptualization, S.G.; Literature Review, S.G.; Methodology, S.G., E.J., D.A., and B.F.N.; Formal analysis, S.G., D.A., E.J., and B.F.N.; Writing—original draft preparation, S.G., E.J., D.A.; Writing—review and editing, S.G., D.A., E.J., and B.F.N. All authors have read and agreed to the published version of the manuscript.

**Funding:** This research received no external funding. D. A acknowledges the part of the work that is based on experience in the +CityxChange (Positive City ExChange) project, funded by the European Union's Horizon 2020 research and innovation programme under Grant Agreement No. 824260.

**Acknowledgments:** We thank the different project members from the case studies who provided insight for our research. We would like to express our thanks to the reviewers and the academic editors for their guidance, time, and input that they have made, and their contributions to raising the overall quality of the paper.

**Conflicts of Interest:** The authors declare no conflict of interest.

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
