# Peer review of "The Governance Approach of Smart City Initiatives. Evidence from Trondheim, Bergen, and Bodø†"

_infrastructures, doi:10.3390/infrastructures5040031_

Round 1

Reviewer 1 Report

This article brings up a very interesting subject, which is the use of governance in smart cities.

The changes were adequate and the bibliographic research expanded. The study was very good.

Reviewer 2 Report

I really appreciate the author(s) efforts to revise the manuscript as per the suggestions given in the last version of the manuscript. The manuscript is improved a lot and now clearly presents the idea.

Reviewer 3 Report

Authors addressed most of my comments, the manuscript can be accepted for publication. 

This manuscript is a resubmission of an earlier submission. The following is a list of the peer review reports and author responses from that submission.

Round 1

Reviewer 1 Report

Dear authors, the proposed article is a good starting point for the discussion of governance that is relevant and current in relation to Smart Cities, but needs analysis to support it. Following are points to improve:

Point 1: References need updating. Very relevant articles have been developed in the last five years and disregarded by the authors, the bibliographic reference should be expanded and a scientific research methodology used.

Point 2: There is already a consensus of the authors that smart cities include digital and sustainable cities, the concepts have been unified in recent years. Lines 34 and 35.

Point 3: The modern concept of smart cities already encompasses the concept of sustainable cities. Lines 90 and 91. This understanding is confirmed in lines 98 and 99; 190 and 191.

Point 4: The introduction needs to be redone, as the text gives a conceptualization of the governance of network actors without demonstrating any analysis methodology up to line 231.

Point 5: Although the text quotes the term “governance” a lot, the authors could not translate it into a management model for smart cities.

Point 6: The methodology needs to be improved, there is no concern of researchers to demonstrate which concepts of the methodology used, which qualitative or quantitative analysis parameters are used to promote the results, even partial.

Point 7: The applied human sciences analysis methodology is not appropriate to the present article, and the authors should consider using a governance measurement matrix, based on a previously validated model.

Point 8: From the beginning to the end of the methodology there is no scientific reference to corroborate the proposal, it needs to be redone. Line 231 to 352.

Point 9: There is no conceptualization in the text of urban planning, sustainable development and the key factors of a Smart City.

Point 10: Conclusions cannot be made without a sound scientific basis.

Reviewer 2 Report

The paper lacks the discussion about the aim of the study, the primary objective is not clear, significance and the comparison with the state of the art. The introduction is quite verbose and does not cover the motivation and the contribution of the work. It needs proper extensions.  It should state the motivation of the authors to conduct the present work and the way that it could be assistive to specific applications and systems. Background topics and related approaches are mainly illustrated in the Introduction. However, a better comparison with the author's approach should be made in the discussion or after the discussion section. One of the limitations is the lack of discussion of the state of the art. The related work needs to be presented in better detail each one of the works addressed describing its aims (research questions specified) and the results collected. Please explain which authors’ work face the gaps in the literature. One more important concern is the lack of discussion on the application of the current technique. Moreover, no compression what’s so ever has been made with the state of the art? The conclusion section needs to be re-written to better present the study. Finally, extensive editing of the English language and style required. The grammar needs polishing. Please have the paper proofread again. In the lights of the above-mentioned issues, the manuscript in the current form lack novelty, research contribution, and application. So it needs to be seriously rewritten and proofread.

Reviewer 3 Report

presentation could be improved by introduction of diagrams and figures,

Reviewer 4 Report

The manuscript focuses on an important topic to international academic and public and private sector audiences and relevant to the journal. Where in general the manuscript has some merits, as it was initially prepared as a conference presentation, it need some major consolidations to be considered for publication in this journal.

1) The manuscript promises to conceptualise the governance of smart city development but actually fails to do so. I would suggest tone down the aim to perhaps generating insights into conceptualisation the governance of smart city development.

2) The manuscript also does not provide an adequate background coverage of the scholarly work on smart cities at the Introduction and the second section dedicated to the literature. The following articles, along with others, could be useful building a stronger literature background.

Yigitcanlar, T., Kamruzzaman, M., Foth, M., Sabatini-Marques, J., da Costa, E., & Ioppolo, G. (2019). Can cities become smart without being sustainable? A systematic review of the literature. Sustainable cities and society45, 348-365. https://doi.org/10.1016/j.scs.2018.11.033 

3) The methodology section is not clear. Provide a more to the point section on what the methodological approach is, how the study is designed, how the data is collected, how it is analyses, and so on.

4) It is not clear how the case studies are investigated, this also links with the methodological ambiguity. How the result tables 1-3 are developed? How you have reached to the presented findings?

5) Lastly the conclusion section does not address the so-what question. 

6) While the manuscript has the potential, due to the abovementioned major issues in its present form it lacks of originality, novelty and significance of the findings. Without a sound theoretical and methodological background, and clear findings, the research study reported in the manuscript has inadequate impact to the field and practice.

Good luck at the revision.